# Beneficial Effect of ACI-24 Vaccination on Aβ Plaque Pathology and Microglial Phenotypes in an Amyloidosis Mouse Model

**DOI:** 10.3390/cells12010079

**Published:** 2022-12-24

**Authors:** Jasenka Rudan Njavro, Marija Vukicevic, Emma Fiorini, Lina Dinkel, Stephan A. Müller, Anna Berghofer, Chiara Bordier, Stanislav Kozlov, Annett Halle, Katrin Buschmann, Anja Capell, Camilla Giudici, Michael Willem, Regina Feederle, Stefan F. Lichtenthaler, Chiara Babolin, Paolo Montanari, Andrea Pfeifer, Marie Kosco-Vilbois, Sabina Tahirovic

**Affiliations:** 1German Center for Neurodegenerative Diseases (DZNE), 81377 Munich, Germany; 2AC Immune SA, 1015 Lausanne, Switzerland; 3Neuroproteomics, School of Medicine, Klinikum rechts der Isar, Technical University of Munich, 80333 Munich, Germany; 4German Center for Neurodegenerative Diseases (DZNE), 53127 Bonn, Germany; 5Biomedical Center (BMC), Ludwig-Maximilians University Munich, 80539 Munich, Germany; 6Monoclonal Antibody Core Facility, Helmholtz Zentrum München, German Research Center for Environmental Health (GmbH), 85764 Neuherberg, Germany; 7Munich Cluster for Systems Neurology (SyNergy), 81377 Munich, Germany

**Keywords:** Alzheimer’s disease, immunotherapy, microglia, Aβ vaccine, ACI-24

## Abstract

Amyloid-β (Aβ) deposition is an initiating factor in Alzheimer’s disease (AD). Microglia are the brain immune cells that surround and phagocytose Aβ plaques, but their phagocytic capacity declines in AD. This is in agreement with studies that associate AD risk loci with genes regulating the phagocytic function of immune cells. Immunotherapies are currently pursued as strategies against AD and there are increased efforts to understand the role of the immune system in ameliorating AD pathology. Here, we evaluated the effect of the Aβ targeting ACI-24 vaccine in reducing AD pathology in an amyloidosis mouse model. ACI-24 vaccination elicited a robust and sustained antibody response in APPPS1 mice with an accompanying reduction of Aβ plaque load, Aβ plaque-associated ApoE and dystrophic neurites as compared to non-vaccinated controls. Furthermore, an increased number of NLRP3-positive plaque-associated microglia was observed following ACI-24 vaccination. In contrast to this local microglial activation at Aβ plaques, we observed a more ramified morphology of Aβ plaque-distant microglia compared to non-vaccinated controls. Accordingly, bulk transcriptomic analysis revealed a trend towards the reduced expression of several disease-associated microglia (DAM) signatures that is in line with the reduced Aβ plaque load triggered by ACI-24 vaccination. Our study demonstrates that administration of the Aβ targeting vaccine ACI-24 reduces AD pathology, suggesting its use as a safe and cost-effective AD therapeutic intervention.

## 1. Introduction

Accumulation of amyloid-β (Aβ) is hypothesised to be an early event in a complex neurodegenerative cascade that leads to cognitive and functional impairments in Alzheimer´s disease (AD) [1,2]. There are multiple lines of evidence for an imbalance between production and clearance of Aβ as the initiating factor in AD pathology. Therefore, Aβ has emerged as the most extensively pursued therapeutic target [3]. Microglia are immune cells in the brain that surround and phagocytose Aβ plaques [4,5,6]. Interestingly, during aging and in AD, their phagocytic capacity declines [7,8,9,10]. The importance of microglial phagocytic capacity is emphasised in a number of genome-wide association studies that identified different microglial genes associated with increased risk of developing AD [11]. The identified risk genes are functionally linked with defects in endo-lysosomal pathway, phagocytosis and Aβ clearance [12]. These changes are reflected by RNA signatures of microglia under varying pathological conditions classified as homeostatic, ageing and disease-associated [13,14,15,16,17,18]. Furthermore, this is supported by our proteomic analysis of microglia isolated from mouse models of amyloidosis (Microglial Aβ Response Proteins; MARPs) that revealed molecular alterations, correlating with the decline of their phagocytic function [10]. Importantly, microglia in the brain of AD patients bear the potential for repair as reflected by their enhanced Aβ clearance [19]. We have previously shown, using an ex vivo model of AD, that factors secreted by microglia isolated from young (postnatal) mice induced proliferation of aged microglia and reduced Aβ plaque load [9], underscoring the importance of exploring therapeutic strategies that enhance microglial clearance to reduce Aβ burden.

Immunotherapy is a therapeutic strategy gaining clinical validation to alter the disease progression and pathology associated with AD [20]. The ground-breaking approach of immunising AD mouse models with the full-length Aβ sequence (1–42) demonstrated that the development of Aβ plaques could be prevented [21]. Indeed, the first vaccine, AN1792, incorporating the full-length Aβ sequence (1–42), established an early clinical proof of concept [22,23]. However, 6% of patients developed meningoencephalitis [24], a severe side effect attributed to the T cell mediated response against the mid-region of the Aβ sequence (10–24) where the dominant T cell epitope is located [25]. Therefore, next generation vaccines utilised antigens from the N-terminal part of the Aβ sequence to avoid an unwanted autoimmune reaction [26,27]. Together with the active immunisation, passive immunotherapy corroborated the strategy to target Aβ. Aducanumab, a human monoclonal antibody (mAb) that selectively targets Aβ aggregates, provided substantial evidence for efficacy in reducing Aβ [19] and is the first FDA approved drug for the mechanism-based therapy against AD [20]. Recent clinical advancements of several anti-Aβ mAbs further support the feasibility of reducing Aβ burden, despite having difficulties reaching all predefined primary and secondary endpoints [28,29,30,31,32]. Excitingly, a recent press release reported that mAb lecanemab met both primary and secondary endpoints (https://www.bioarctic.se/en/lecanemab-phase-3-clarity-ad-study-in-early-alzheimers-disease-meets-primary-and-all-key-secondary-endpoints-with-high-statistical-significance-5808/; accessed on 8 November 2022), strongly supporting the therapeutic benefits of targeting Aβ. Along these lines, targeting of pre-amyloid seeds can efficiently reduce Alzheimer-like pathology in amyloidosis mice [33], supporting the efficacy of early disease intervention strategies.

Although both active and passive immunisation strategies may show beneficial effects on Aβ pathology, active immunisation offers important advantages. Vaccines are cost-effective, can generate a long-term polyclonal disease modifying response and can be introduced as a prevention strategy [34]. Thus, an active immunisation approach may prove to be more effective in slowing the cognitive decline. Furthermore, the most common safety concern of immunotherapy, the Aβ-related imaging abnormalities (ARIA) reported for several anti-Aβ monoclonal antibodies [19,28,30,32,35], seem to be less frequent for Aβ targeting vaccines [36,37,38,39,40].

It is, therefore, important to generate new vaccines with the potential to reduce Aβ and improve cognitive decline. ACI-24 is a liposomal vaccine that anchors the Aβ sequence (1–15) between palmitoylated lysine tandems, thereby adopting the aggregating beta-sheet structure [41]. Pre-clinical studies using ACI-24 showed that vaccination prevents memory defects in mouse models of amyloidosis [41] and Down syndrome [42]. Furthermore, the produced antibodies were more specific for aggregated Aβ, including oligomers, with minor reactivity to monomers [41,43], underscoring the specificity of this vaccine towards pathological forms of Aβ. Recently, an optimised formulation of ACI-24 has been developed, harbouring additional non-target T cell epitopes. This modification improves the immune response and triggers high titres against the neurotoxic and pathological species pyroglutamate Aβ3–42 [44,45,46].

We show here that the efficacy of the ACI-24 vaccine is linked to the severity of Aβ deposition. Although ACI-24 elicited a substantial immune response in all vaccinated APPPS1 [47] animals, AD pathology, including Aβ plaque load, Aβ plaque associated ApoE and dystrophic neurites, could be more robustly reduced when vaccination was initiated at stages with less severe Aβ deposition. Furthermore, we observed an increased proportion of inflammasome component NLRP3-positive plaque-associated microglia following ACI-24 vaccination. In contrast to this local microglial activation at Aβ plaques, globally, we observed a more ramified morphology of Aβ plaque-distant microglia, supporting beneficial effects of ACI-24 vaccination on microglial phenotypes. Accordingly, bulk transcriptomic analysis revealed a trend towards reduction in the expression of several disease-associated microglia (DAM) signatures that is in line with reduced Aβ plaque load triggered by ACI-24 vaccination.

Taken together, this in vivo study shows the strength of the ACI-24 vaccine in triggering robust immune responses and effectively reducing AD pathology. Our work supports the concept of using a vaccine approach as a safe and cost-effective AD therapeutic intervention and initiating preventive trials at early stages of Aβ plaque deposition.

## 2. Materials and Methods

### 2.1. Animals

Female mice from the hemizygous APPPS1 mouse line overexpressing human APPKM670/671NL and PS1L166P under the Thy-1 promoter [47] were used in this study. Mice were bred on a wild-type (WT) C57BL/6J background with ad libitum access to water and standard mouse chow (Ssniff Ms-H, Ssniff Spezialdiaeten GmbH, Soest, Germany). ApoE knock-out mice (B6.129P2-Apoetm1Unc/J) [48] were used for the generation of mouse ApoE antibodies. Animals were kept under a 12/12 h light–dark cycle. All animal experiments were carried out in accordance with the German animal welfare law and have been approved by the government of Upper Bavaria (ROB-55.2-2532.Vet_02-17-153 and ROB-55.2-2532.Vet_03-17-68).

### 2.2. Animal Treatment

Six-week-old female APPPS1 littermates were subcutaneously injected with 200 μL of ACI-24 (80 μg per dose of antigen) [41] or PBS (10010056, Gibco) as control. Mice were vaccinated every 2 weeks until 10 weeks of age (in total 3 doses of the vaccine) as schematically outlined in Figure 1A. At the age of 12 weeks, mice were sacrificed, re-genotyped and brains and plasma were collected for further analysis. The cohort size was 10 animals per condition (*n* = 10) for the immunohistochemical and biochemical analysis. The control group consisted of 10 female APPPS1 animals (5 were offspring of a transgenic father and WT mother, and 5 were offspring of a transgenic mother and WT father). The ACI-24 treated group included 10 female animals (4 transgenic father and 6 transgenic mother offspring). The proteomic cohort consisted of 5 control (3 transgenic father and 2 transgenic mother offspring) and 6 ACI-24 treated animals (3 transgenic father and 3 transgenic mother offspring).

### 2.3. Blood Collection

Blood was collected before the first immunisation (baseline), and at one, three and six weeks after the first immunisation (Figure 1A). Blood samples were collected from the facial vein using animal lancet (GR 5 MM, Braintree Scientific, Braintree, MA, USA) or by a heart puncture at the end of the study. Samples were collected in heparin tubes (16.443, Sarstedt, Nümbrecht, Germany), mixed up and down by inverting the tube at least 4 times, and kept on ice until further processing. Blood was then centrifuged at 17,000× *g* for 10 min at 4 °C to collect the plasma. Supernatant was aliquoted and stored at −80 °C until analysis.

### 2.4. Quantification of Antigen-Specific Antibodies

Specific anti-Aβ1-42 IgG was measured by ELISA as previously described [46]. Briefly, plates were coated with 10 μg/mL Aβ1-42 peptide film (Bachem, Bubendorf, Switzerland) overnight at 4 °C. After washing the plate with PBS/0.05% Tween 20 and blocking with 1% BSA/PBS/0.05% Tween 20 for 1 h at 37 °C, the plasma was added in serial dilutions (1:100, 1:200, 1:400, 1:800, 1:1600, 1:3200, 1:6400 and 1:12,800) and incubated for 2 h at 37 °C. The 6E10 antibody (dilution 1:1600, 803002, Biolegend, Amsterdam, Netherlands) was used as a standard and 4G8 (dilution 1:5000, 800702, Biolegend, Amsterdam, Netherlands) as a positive control. After washing, plates were incubated with the detection antibody, an alkaline phosphatase conjugated anti-mouse IgG (dilution 1:7000, 115-055-164, Jackson ImmunoResearch Europe, Ely, UK) for 2 h at 37 °C. After washing, plates were incubated with the phosphatase substrate pNPP (S0942, Sigma-Aldrich, Merck, Taufkirchen, Germany) and read at 405 nm by using an ELISA plate reader. Results are expressed with reference to the serial dilution of the 6E10 antibody.

### 2.5. Generation of the ApoE Antibody 26C11

A peptide comprising amino acids _20_GEPEVTDQLEWQSN_33_ of mouse ApoE protein was synthesised and coupled to OVA (Peps4LS, Heidelberg, Germany). ApoE knock-out mice were immunised with a mixture of 40 µg peptide, 6 nmol CpG oligonucleotide 1668 (Tib Molbiol, Berlin, Germany) in 200 µL PBS and 200 µL incomplete Freund’s adjuvant. A boost without adjuvant was given 14 weeks after the primary injection. Fusion of mouse spleen cells with the myeloma cell line P3X63-Ag8.653 (ATCC; CRL-1580) was performed using standard procedures. Hybridoma supernatants were tested in a flow cytometry assay (iQue, Intellicyt; Sartorius, Göttingen, Germany) for binding to biotinylated ApoE peptide coupled to streptavidin beads (PolyAN, Berlin, Germany). Using Atto-488-coupled isotype-specific monoclonal rat-anti-mouse IgG secondary antibodies, antibody binding was analysed using ForeCyt software v9.0 (Sartorius, Goettingen, Germany). Positive supernatants were further analysed by Western blot analysis and selected hybridoma cells were subcloned by limiting dilution to obtain stable monoclonal cell lines. Experiments in this work were performed with hybridoma supernatant of clone APO1M 26C11 (mouse IgG2b/ƙ).

### 2.6. Immunohistochemistry

The brain was perfused with 0.9% NaCl isotonic solution for 5 min (B. Braun, Melsungen, Germany) and the right hemisphere was post-fixed in 4% paraformaldehyde in 0.1 M PBS for 6 h. Hemispheres were cryoprotected (30% sucrose in 0.1 M PBS), embedded in Optimal Cutting Temperature medium (4583, Science Services, München, Germany) on dry ice and stored at −80 °C until sectioning. Sagittal sections (30 μm thick) were cut using a cryostat (CryoSTAR NX70, Thermo Fisher Scientific, Schwerte, Germany) and placed in 0.1 M PBS for direct staining. Alternatively, sections were stored in anti-freezing solution (30% glycerol, 30% ethylenglycol, 10% 0.1 M PBS, pH 7.2–7.4 and 30% dH_2_O) at −20 °C and washed in 0.1 M PBS before staining. For ApoE antibody, antigen retrieval step was carried out before permeabilization. Sections were incubated in pre-warmed 10 mM citric acid buffer pH 6 for 20 min at 96 °C with shaking. Afterwards, sections were washed with PBS. When using primary mouse antibodies, sections were permeabilized in PBS/0.5% Triton X-100 for 30 min, followed by 1 h incubation in FAB fragment at room temperature to block for unspecific epitopes (1:100 in PBS, 715-007-003, Jackson ImmunoResearch Europe, Ely, UK) and washed in PBS/0.2% Triton X-100 before blocking. For all the other primary antibodies, sections were directly permeabilized and blocked in PBS/0.5% Triton X-100/5% normal goat or donkey serum for 1 h at room temperature and incubated overnight at 4 °C in primary antibody diluted in blocking solution. After washing with PBS/0.2% Triton X-100, sections were incubated in appropriate fluorophore-conjugated secondary antibodies (1:500, Life Technologies) together with nuclear stain Hoechst 33342 (1:2000, H3570, Thermo Fisher Scientific, Schwerte, Germany) and dense fibrillar plaque core staining Thiazin red (1:1000 dilution, 12648, Morphisto, Offenbach am Main, Germany) or Methoxy-X04 (1:10,000, ab142818, Abcam, Cambridge, UK) for 2 h at room temperature. After washing, sections were mounted onto glass slides (Thermo Fisher Scientific, Schwerte, Germany), dried for 1 h, mounted using Gel Aqua Mount media (F4680, Sigma-Aldrich, Merck, Taufkirchen, Germany) and analysed by confocal microscopy.

Primary antibodies used: Aβ 3552 (0.74 µg/mL) [49], Aβ NAB228 (1:500, sc-32277, Santa Cruz Biotechnology, Heidelberg, Germany) [10], BACE1 (1:400, ab108394, Abcam, Cambridge, UK), ApoE (1:200, hybridoma supernatant, clone 26C11), IBA1 (1:500, ab5076, Abcam, Cambridge, UK), IBA1 (1:500, 234308, Synaptic Systems, Göttingen, Germany), NLRP3 (1:50, BSS-BS-10021R-100, Biozol Diagnostics Vertrieb, Eching, Germany) and CD68 (1:500, MCA1957GA, Bio-Rad, Feldkirchen, Germany).

### 2.7. Image Acquisition, Analysis and Quantification

All quantifications were performed by investigators blinded to experimental conditions. For the quantification of total Aβ (NAB288 antibody), fibrillar Aβ (Thiazin red staining), ApoE and BACE1, 16 images per brain were systematically taken from comparable neocortical regions using a confocal microscope (Zeiss LSM 900). Quantification was performed using a self-programmed macro with ImageJ software v.1.53c.

For the microglial recruitment analysis and CD68 coverage, randomly selected plaques from the neocortical region were imaged (30 images per brain) as described previously [10]. Within the same experiment, microscopy acquisition settings were kept constant. For every image, maximal intensity projection was created from every z-stack using ImageJ software v.1.53c. A defined region of interest (ROI) was manually drawn around every plaque and measured for the Aβ and CD68 coverage area. The number of microglia (Iba1 + cells) recruited to the plaque area was counted within the ROI and through the z-stack. Total CD68 coverage area was normalised to the number of microglia (Iba1+ cells) recruited to the plaque within the ROI. The number of microglial cells at Aβ plaques was normalised to the area covered by Aβ (3552 antibody) and expressed as number of microglial cells per μm^2^ of Aβ plaque.

For the characterisation of NLRP3 activation in plaque-associated microglia, randomly selected confocal images (Z-stack 25 µm; slice distance (Z) 0.5 µm) from the neocortical region were acquired with a 40× objective (8 images per animal, each containing 1–5 plaques). Images were analysed using Imaris (Imaris 9.2.1). The total number of cells positive for NLRP3 was quantified manually and normalised to the total number of plaque-associated microglia.

For the microglial morphology analysis, randomly selected images from the neocortical region were acquired with a 40× objective at a resolution of 1024 × 1024 pixels and a slice distance (Z) of 0.4 µm using the maximum possible number of Z-stacks. Analysis was carried out manually using ImageJ as previously described [50]. Briefly, single microglia (IBA1 positive cells) that were clearly distinguishable from one another and at least 1.5 plaque diameter away were selected through the Z-stack (8–14 microglia per brain). The threshold was adjusted, and the noise was removed from the binary image. The cells were further analysed with the plugin Sholl analysis 4.1.1 by measuring the maximum radius of the cell soma, the radius of the longest branch of the cell and the number of primary branches. Schoenen ramification index (RI) was automatically calculated (number of end branches/number of primary branches) and collected for each cell. Area and perimeter of the cell were also measured to calculate the circularity index (4π[area]/[perimeter]^2^).

### 2.8. Biochemical Analysis of the Brain

The brain was perfused with 0.9% NaCl isotonic solution for 5 min (B. Braun, Melsungen, Germany) and the left hemisphere (without olfactory bulb, brain stem and cerebellum) was snap frozen and subsequently pulverised using CP02 cryoPREP Automated Dry Pulverizer (Covaris, Brighton, UK) and liquid nitrogen. Pulverised hemispheres were stored at −80 °C until processing. Aliquots of brain powder were lysed on ice for 30 min in lysis buffer (150 mM NaCl, 50 mM Tris pH 7.5, 1% Triton X-100) supplemented with protease and phosphatase inhibitor cocktail (Roche). Samples were then centrifuged at 17,000× *g* for 30 min at 4 °C and supernatants were collected (Triton X-100 soluble faction). Pellets were further resuspended in 70% formic acid (FA), sonicated for 5 min and centrifuged at 186,000× *g* for 30 min at 4 °C. FA supernatants (formic acid soluble fraction) were then diluted 1:20 in neutralisation buffer (1 M Tris base, pH 9.5).

For Western blot analysis, the protein content from Triton X-100 soluble fraction was quantified using BC assay (UP40840A, Interchim, Montluçon, France) according to manufacturer’s protocol with a NanoQuant Infinite M200 Pro. A total of 10 to 20 μg of protein was loaded on a Novex 10–20% tricine protein gel (EC66252BOX, Thermo Fisher Scientific, Schwerte, Germany), followed by blotting on nitrocellulose membrane (Merck, Taufkirchen, Germany) using anti-Aβ 2D8 (1:50) [49] and anti-β-actin (1:1000, A5316, Sigma-Aldrich, Merck, Taufkirchen, Germany) as a loading control. Blots were developed using horseradish peroxidase-conjugated secondary antibodies (Promega, Walldorf, Germany) and the ECL chemiluminescence system (Amersham Biosciences Europe, Freiburg im Breisgau, Germany) with an ImageQuant LAS 4000 series.

ELISA analysis of Aβ was carried out according to the manufacturer´s protocol (K15200G, MSD, Rockville, MD, USA). A total of 10 μg of protein from the Triton X-100 soluble fraction was loaded, while the formic acid soluble fraction was loaded according to the protein amount of the Triton X-100 soluble fraction (diluted at least in 1 to 5 ratio).

### 2.9. Brain Gene Expression Profiling

An aliquot of 20 to 30 mg of the pulverised hemispheres (without olfactory bulb, brain stem and cerebellum) was taken for the total RNA extraction. Total RNA was isolated using the RNeasy Mini kit (74104, Qiagen, Hilden, Germany) and the quality of the sample was assessed using the Agilent RNA 6000 Nano Kit (5067-1511, Agilent technologies, Waldbronn, Germany) according to the manufacturer´s instruction. A total of 100 ng RNA per sample was subjected to gene expression profiling using the nCounter^®^ Neuropathology panel and Glial Profiling panel from NanoString (NanoString Technologies, Seattle, WA, USA). Data were analysed using the nSolver Analysis Software, version 4.0. The background of all the samples was subtracted using a threshold with a defined value of 20. Thereafter, gene expression levels in each sample were normalised against the geometric mean of positive controls and housekeeping genes (provided by the panel; excluding genes with CV higher than 20% from the lowest sample and excluding genes with the average count less than 100).

### 2.10. Isolation of Primary Microglia

After the brain had been removed from the skull, the olfactory bulb, brain stem and cerebellum were removed, meninges cleaned and primary microglia were acutely isolated (without culturing) from the remaining cerebrum using MACS technology (Miltenyi Biotec, Bergisch Gladbach, Germany) according to manufacturer’s instructions and as previously described, with some modification [51]. Samples were dissociated, first by enzymatic digestion using papain (200 U, P3125, Sigma-Aldrich, Merck, Taufkirchen, Germany) and afterwards with mechanical dissociation using three fire-polished glass Pasteur pipettes of decreasing diameter. CD11b+ microglia were magnetically separated (Miltenyi Biotec, Bergisch Gladbach, Germany) to obtain CD11b-enriched (microglia-enriched) and CD11b-depleted (microglia-depleted) fractions. After two washes with HBSS (Gibco, supplemented with 7 mM HEPES), microglia-enriched pellets were frozen in liquid nitrogen and stored at −80 °C until processing.

### 2.11. Sample Preparation for Proteomics

Microglia-enriched pellets from 5 APPPS1 mice (3 transgenic father and 2 transgenic mother offspring) treated with PBS and 6 APPPS1 mice (3 transgenic father and 3 transgenic mother offspring) treated with ACI-24 were subjected to proteomic analysis. The cell pellets were lysed in 200 µL of STET lysis buffer (50 mM Tris, 150 mM NaCl, 2 mM EDTA, 1% Triton, pH 7.5) at 4 °C with intermediate vortexing. The samples were centrifuged for 5 min at 16,000× *g* at 4 °C to remove cell debris and undissolved material. The supernatant was transferred to a protein LoBind tube (Eppendorf) and the protein concentration estimated using the Pierce 660 nm protein assay (Thermo Fisher Scientific, Schwerte, Germany). A protein amount of 15 µg per sample was subjected to tryptic digestion. First, 100 mM MgCl_2_ was added to a final concentration of 10 mM and DNA was digested with 25 units Benzonase (Sigma-Aldrich, Merck, St. Louis, MO, USA) for 30 min at 37 °C. Proteins were reduced at 37 °C for 30 min with 15 mM dithiothreitol (DTT) followed by cysteine alkylation with 60 mM iodoacetamide (IAA) for 30 min at 20 °C. Excess of IAA was removed by adding DTT. Detergent removal and subsequent digestion with 0.2 µg LysC and 0.2 µg trypsin (Promega, Walldorf, Germany) were performed using the single-pot, solid-phase-enhanced sample preparation as previously described [52]. After vacuum centrifugation, peptides were dissolved in 20 µL of 0.1% formic acid (Biosolve Chimie, Dieuze, France) and indexed retention time peptides were added (iRT Kit, Biognosys, Schlieren, Switzerland).

### 2.12. LC-MS/MS Analysis

The LC-MS/MS analyses were performed on a nanoElute system (Bruker Daltonics, Bremen, Germany) which was online coupled with a timsTOF pro mass spectrometer (Bruker Daltonics, Bremen, Germany). An amount of 400 ng of peptides per sample were separated on a self-packed 15 cm C18 column (75 µm ID) packed with ReproSil-Pur 120 C18-AQ resin (1.9 µm, Dr. Maisch GmbH, Ammerbuch, Germany). For peptide separation, a binary gradient of water and acetonitrile (B) with a length of 120 min was applied at a flow rate of 300 nL/min and a column temperature of 50 °C (0 min: 2% B; 2 min: 5% B; 94 min: 24% B; 112 min: 35% B; 120 min: 60% B).

A Data Independent Acquisition Parallel Accumulation Serial Fragmentation (DIA-PASEF) method was used. One MS1 full scan was followed by 32 sequential DIA windows with 26 *m*/*z* width for peptide fragment ion spectra with an overlap of 1 *m*/*z* covering a scan range of 400 to 1201 *m*/*z*. The ramp time was fixed to 100 ms and 2 windows were scanned per ramp. This resulted in a total cycle time of 1.8 s.

The software DIA-NN version 1.8 was used to analyse the data [53]. The raw data were searched against a one protein per gene database from *Mus musculus* (UniProt, 21966 entries, download: 09-04-2021) using a library free search. Trypsin was defined as protease and 2 missed cleavages were allowed. Oxidation of methionines and acetylation of protein N-termini were defined as variable modifications, whereas carbamidomethylation of cysteines was defined as fixed modification. The precursor and fragment ion *m*/*z* ranges were limited from 400 to 1201 and 200 to 1700, respectively. An FDR threshold of 1% was applied for peptide and protein identifications. The mass accuracy and ion mobility windows were automatically adjusted by the software. The match between runs option was enabled.

Generated label-free quantification (LFQ) outputs were log2 transformed and an average log2 fold change was calculated for each protein, which was identified with at least 2 unique peptides in at least 2 biological replicates per condition. Changes in protein abundance were evaluated using a Student’s *t*-test between the log2 LFQ intensities of the two experimental groups. A permutation-based FDR estimation was used to account for multiple hypotheses (*p* = 5%; s0 = 0.1) using the software Perseus 1.6.14.0. Volcanos only display proteins, which were identified often enough to apply statistical tests [54,55].

## 3. Results

### 3.1. ACI-24 Triggers Substantial Immune Response

To investigate the effect of the ACI-24 vaccine, we performed an in vivo study using a fast-progressing amyloidosis mouse model APPPS1 [47]. Aβ deposition in this mouse model is first detectable in the cortex at the age of six to eight weeks [47]. ACI-24 vaccination was initiated at the age of six weeks, thus together with the onset of the Aβ deposition. The vaccination paradigm included three doses of the vaccine or PBS control applied every two weeks and the blood was collected at baseline, between each vaccination and at the end of the study as outlined (Figure 1A). To avoid sex-specific differences in Aβ pathology, only female animals were included into the study. The study ended at 12 weeks of age, when the Aβ pathology is already abundant throughout the cortex [10,47]. The ACI-24 vaccine triggered a robust immune response as measured by the anti-Aβ42 IgG antibody titres detectable in the plasma (Figure 1B). The anti-Aβ42 IgG antibody titres were observed seven days after the first vaccination and sustained throughout the study. We observed comparable anti-Aβ42 total IgG levels following the vaccination of APPPS1 and WT mice. In contrast with the ACI-24 vaccinated mice, this immune response was not measurable in the mice injected with PBS as a control.

### 3.2. ACI-24 Reduces Aβ Load in APPPS1 Mice

To analyse the effects of the ACI-24 on Aβ pathology, we performed an immunohistochemical analysis of cortical Aβ deposition. ACI-24 vaccination had significant effects on the plaque pathology (Figure 1C). We observed a significant reduction in the Aβ plaque coverage using the anti-Aβ antibody NAB228 (Figure 1C,D). Additionally, the Aβ plaque number showed trends to reduction following vaccination (Figure 1E). In agreement with the reduction of the Aβ coverage, the Aβ plaque size was also significantly decreased in the ACI-24 vaccinated mice (Figure 1F). Surprisingly, we identified two subpopulations of analysed female mice that had clearly visible differences in the Aβ plaque load, regardless of the vaccination. The mice that had higher Aβ plaque pathology were the offspring of a transgenic father and a WT mother (Figure 1D–F, marked in blue circles), while the mice with the lower Aβ plaque burden were animals bred with the opposite mating scheme—transgenic mother and a WT father (Figure 1D–F, marked in pink circles). When we separated the two subgroups of the control treated mice, there was a significantly higher coverage and number, and a trend towards increased size of Aβ plaques, in the offspring generated by the transgenic father compared to the transgenic mother (Appendix A). Thus, the transgenic father offspring developed a more pronounced Aβ pathology, compared to the transgenic mother offspring.

Taken together, ACI-24 shows a downregulation of coverage and size of the Aβ plaques in all vaccinated animals. Notably, we observed differences in the Aβ plaque load depending on which parent is carrying the APPPS1 transgene, providing us with the rational for the observed experimental heterogeneity.

### 3.3. ACI-24 More Effectively Reduces Aβ Pathology in the Offspring Generated by the Transgenic Mother

Since we observed differences in the Aβ plaque load depending on which parent is carrying the APPPS1 transgene, we next separately analysed the efficacy of ACI-24 vaccination according to the mating scheme. The transgenic father offspring had a minor effect on Aβ plaque pathology following ACI-24 vaccination (Appendix A). In contrast, the analysis of the transgenic mother offspring revealed a significantly reduced Aβ coverage, plaque number and plaque size following ACI-24 vaccination (Figure 2A–C and Appendix A). In line with these results, we observed a significant reduction in the coverage and number of fibrillar Aβ plaques as well as trends to reduced size of the fibrillar Aβ plaques (Figure 2D–F and Appendix A). Next, we analysed levels of the Triton X-100 extractable (soluble) and formic acid extractable (insoluble) Aβ in the cerebrum of the vaccinated mice via Western blotting (Figure 2G). Transgenic mother offspring had a significant decrease in the Aβ in the Triton X-100 and formic acid soluble fractions following ACI-24 vaccination (Figure 2H,I). To investigate different Aβ species, we performed MSD ELISA of soluble and insoluble Aβ and observed a trend towards reduction of both Aβ40 and Aβ42 species in both fractions of the transgenic mother offspring (Figure 2J,K), supporting the immunohistochemical and Western blot data.

Overall, we demonstrate a more consistent and robust reduction of the Aβ pathology following ACI-24 vaccination in the offspring generated by the transgenic mother, supporting the hypothesis that lower Aβ load is of advantage for the vaccination efficacy.

### 3.4. Microglia at the Plaque Are Changed Following ACI-24 Vaccination

To analyse if ACI-24 vaccination affected microglial phenotypes, we investigated microglial recruitment to Aβ plaques (Figure 3A,B). We observed a trend towards an increased number of microglia at the plaques (Figure 3A and Appendix A) and a trend towards increased CD68 coverage (Figure 3B and Appendix A) following ACI-24 vaccination (transgenic mother offspring). Notably, there was a significant increase in the proportion of NLRP3 positive microglia at the plaque in the ACI-24 vaccinated mice (transgenic mother offspring) (Figure 3C,D), suggesting a local activation of plaque-associated microglia that may contribute to Aβ plaque reduction. However, bulk proteomic analysis of acutely isolated microglia from both transgenic mother and father offspring revealed no major changes of microglial phenotypes following ACI-24 vaccination (Appendix A). Local changes may have not been captured by the bulk proteomic analysis, or may be less pronounced as both transgenic father and mother animals were included into the proteomic analysis. Overall, this suggests that the vaccine-mediated reduction in Aβ deposition may be a valid therapeutic intervention, as there are no excessive changes in microglial phenotypes. This observation is of particular relevance as excessive overactivation of microglia often leads to unwanted side effects that may compromise the therapeutic benefits.

### 3.5. Global Beneficial Microglial Effect of the ACI-24 Vaccination

In addition to bulk proteomic analysis of microglia, we also performed a bulk transcriptomic analysis of cerebrum (transgenic mother offspring) to exclude possible gene expression changes in other brain cells following ACI-24 vaccination. To this end, we analysed the expression of genes with the nCounter Glial Profiling Panel (Figure 4A and Appendix A) and the Neuropathology Panel (Figure 4B and Appendix A) (NanoString Technologies, Seattle, WA, USA) in the cerebrum of the vaccinated mice. The Glial Profiling Panel addresses the crosstalk between glial cells, peripheral immune cells and neurons by assessing cell stress and damage responses, inflammation, peripheral immune invasion, neurotransmission, glial cell homeostasis and activation. The Neuropathology Panel covers the main aspects of neurodegeneration. Expression levels of 757 (Glial Profiling Panel) or 760 (Neuropathology Panel) genes were normalised against the geometric mean of 12 (Glial Profiling Panel) or 9 (Neuropathology Panel) housekeeping genes. Using this comprehensive analysis, we detect only minor changes in bulk gene expression profiles, supporting the notion that ACI-24 vaccination does not trigger overt changes in gene expression profiles of brain cells (Figure 4A,B, Appendix A). Among the few genes that were differentially expressed following ACI-24 vaccination, we detected a trend towards downregulation of the DAM genes *Trem2* (21%), *Cd68* (17%), *Cd163* (24%) and *Cst7* (55%) (Figure 4A and Appendix A), that is well in line with the beneficial effect of reduced Aβ deposition.

To further support the hypothesis that microglia are globally less activated following ACI-24 vaccination, we analysed the morphology of microglia (transgenic mother offspring) that are not directly in contact with Aβ plaques (plaque-distant microglia) (Figure 5A). Sholl analysis of plaque-distant microglia revealed a trend to the increase in the ramification index in the ACI-24 vaccinated compared to control mice (Figure 5B). Accordingly, the circularity of Iba1+ cells, as well as the area of the cell, was significantly reduced (Figure 5C,D), revealing a more ramified microglial morphology in the ACI-24 vaccinated animals. In line with this, the cell perimeter was significantly increased in the ACI-24 vaccinated mice (Figure 5E).

Together, these data suggest that ACI-24 vaccinated mice have less amoeboid and more ramified morphology of plaque-distant microglia compared to non-vaccinated controls. This morphological analysis is in line with the detected trend in transcriptional reduction of several DAM genes in association with the pronounced reduction of Aβ pathology following ACI-24 vaccination.

### 3.6. ACI-24 Reduces ApoE Protein Levels and Neuronal Injury

To further investigate the effect of vaccination with ACI-24, we assessed plaque related abnormalities such as the levels of ApoE (Figure 6A), the major constituent of Aβ plaques that promotes Aβ aggregation and deposition [56,57,58]. Furthermore, we evaluated BACE1 (Figure 6B), that was previously shown to accumulate in the dystrophic neurites that surround Aβ plaques in APP transgenic mice [59]. The total ApoE coverage (transgenic mother offspring) was significantly reduced following ACI-24 vaccination (Figure 6C and Appendix A). The immunohistochemical analysis of BACE1 revealed reduced BACE1 staining in the ACI-24 vaccinated mice compared to controls (transgenic mother offspring), indicating significant reduction in plaque-associated neuritic dystrophies (Figure 6D and Appendix A). These observations correlate well with the reduction of Aβ plaque load post vaccination.

In summary, these experiments demonstrated that vaccination with ACI-24 elicits a beneficial polyclonal antibody response that acts to diminish Aβ plaques and associated pathologies, including a reduction of ApoE and dystrophic neurites. Furthermore, the mechanism involves the normalisation of microglial morphological and molecular phenotypes, suggesting amelioration of neuroinflammation.

## 4. Discussion

This study assessed the efficacy of targeting Aβ with an active immunotherapy approach, using the ACI-24 vaccine for reducing the Aβ pathology in a progressive amyloidosis mouse model, APPPS1 [47]. ACI-24 immunisation resulted in substantial and sustained anti-Aβ42 antibody titres that were robustly detected in all vaccinated animals. Despite the high intra-group variability (transgenic mother and father offspring), a consistent and significant reduction was observed for the Aβ coverage and Aβ plaque size in the ACI-24 vaccinated versus the control mice. Vaccination efficacy was associated with severity of the Aβ plaque pathology, suggesting that vaccination at earlier disease stages may provide a more favourable therapeutic outcome. Along these lines, the vaccination-mediated effects on the Aβ reduction were more prominently observed in the animals generated by the transgenic mother, which had lower Aβ burden compared to animals generated by the transgenic father.

To exclude the possibility that this unexpected difference in Aβ burden between transgenic mother and father APPPS1 offspring did not happen by chance, this effect should be further validated in a larger animal cohort as our findings may suggest a mechanism of maternal protection. Although the underlying mechanistic insight remains unknown, this protective effect may be mediated by estrogenic action, mitochondrial maternal contribution [60] or the transfer of anti-Aβ antibodies from the mother to the offspring [61]. Indeed, auto anti-Aβ antibodies have been already reported in humans and this was integral for the discovery of therapeutically relevant antibody aducanumab [19]. Moreover, maternal protection and transfer of anti-Aβ antibodies from the mother to the pups was recently described in the 5xFAD model following Aβ vaccination [61].

The robust reduction of the Aβ plaque load triggered by ACI-24 vaccine further translated into the beneficial reduction of dystrophic neurites, i.e., swellings in neurons that reflect abnormal axonal trafficking and accompany Aβ deposition [62,63]. Moreover, there was a reduction in the ApoE levels, a protein known to promote Aβ aggregation and deposition in the plaques [56,57,58].

The reduction of the Aβ plaque load following ACI-24 vaccination was in accordance with globally reduced microglial activation. mRNA analysis revealed a trend towards reduced DAM signatures such as *Trem2*, *Cd68*, *Cd163* and *Cst7* [10,16,18,64]. Accordingly, plaque-distant microglia showed more ramified morphology and reduced circularity following ACI-24 vaccination, suggesting reduced neuroinflammation. A similar effect of ramified microglial morphology following reduction of plaque load was recently observed with targeted deletion of BACE1 in 5xFAD microglia [65]. The beneficial resolution of inflammation may be helpful to reduce side effects such as ARIA that were observed in the clinical trials with anti-Aβ targeting mAbs [66] or Aβ targeting vaccines [36,37,38,39,40]. In contrast to globally reduced microglial activation following ACI-24 vaccination, local microglial activation, assessed by the increase in the proportion of NLRP3-positive plaque-associated microglia, may contribute to reduced Aβ plaque burden [9,67,68]. This is in line with the increased recruitment of microglia to Aβ plaques observed following aducanumab treatment and the proposed mechanism of FcγR-mediated phagocytosis of pathological Aβ [19,69]. Taken together, this regulation of microglial inflammatory phenotypes speaks positively for a safe and successful therapeutic intervention where beneficial microglial responses are feasible without detrimental and uncontrolled microglial overstimulation.

In summary, the key findings of our study are that vaccination with ACI-24 elicits a robust and sustained antibody response that translates into a beneficial effect on plaque load, reducing Aβ plaque-associated ApoE, dystrophic neurites and microglial pathology. Furthermore, these data—albeit generated by using a transgenic animal model—suggest that the efficacy of the vaccination with ACI-24 could be linked to the severity of Aβ deposition, highlighting early and even preventive therapeutic use to modify Aβ plaque deposition.

## Figures and Tables

**Figure 1 cells-12-00079-f001:**
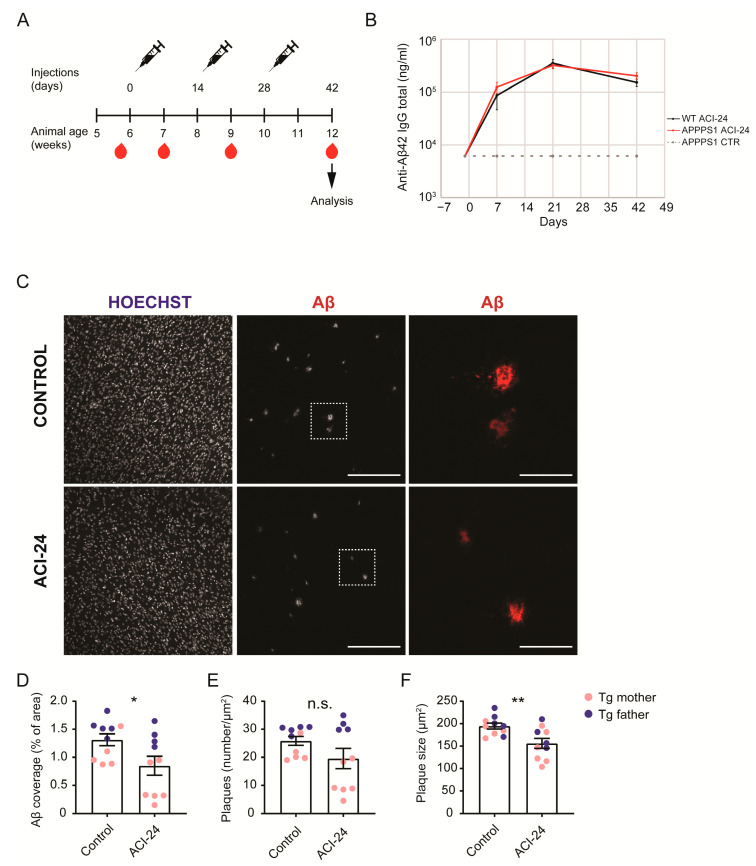
ACI-24 vaccination efficiently reduces Aβ plaque load in APPPS1 mice. Schematic overview of the vaccination paradigm (**A**). ELISA analysis of the anti-Aβ42 total IgG titres (ng/mL) in plasma (**B**). Control APPPS1 mice (PBS treated, in dotted grey) have the baseline titres, while both ACI-24 vaccinated APPPS1 (in red) and ACI-24 vaccinated WT mice (in black) have robust and sustained immune response. Representative images of Aβ (as stained with NAB228 antibody) and nuclei (Hoechst) in cortical sections show decreased Aβ deposition following ACI-24 vaccination. Scale bar for images in the centre column is indicating 200 µm, and scale bar for images in the right column (magnification of dotted boxed regions from centre column images) is indicating 50 µm (**C**). Statistical analysis of 10 control and 10 ACI-24 vaccinated APPPS1 mice for Aβ plaque coverage (**D**), number (**E**) and size (**F**). ACI-24 significantly downregulated Aβ plaque coverage and size. Offspring are from the transgenic mother (pink circles) or father (blue circles). Graphs are presented as mean ± SEM (* *p* < 0.05, ** *p* < 0.001, unpaired two-tailed Student’s *t*-test).

**Figure 2 cells-12-00079-f002:**
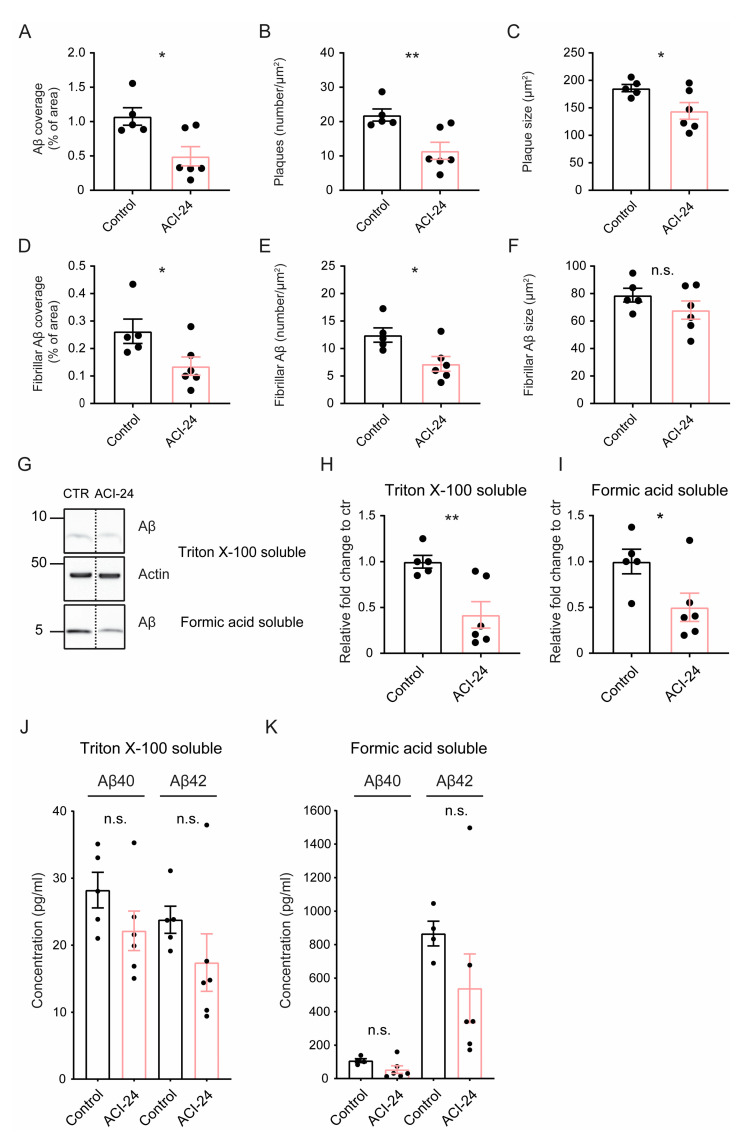
ACI-24 effectively reduces Aβ pathology in the offspring generated by the transgenic mother. Statistical analysis of 5 control and 6 ACI-24 vaccinated mice (transgenic mother offspring) for Aβ plaque coverage (**A**), number (**B**) and size (**C**) and fibrillar Aβ plaque coverage (**D**), number (**E**) and size (**F**). Representative images (**G**) and quantifications (**H**,**I**) of Western blot analysis of Triton X-100 soluble and formic acid soluble Aβ. Of note, representative samples (**G**) were not loaded next to each other, but on the same membrane. Therefore, they are separated by the dashed line. ACI-24 vaccination leads to significant reduction of Aβ in Triton X-100 soluble and formic acid soluble fractions. β-actin represents the loading control. ELISA analysis of Triton X-100 soluble (**J**) and formic acid soluble Aβ (**K**) fractions demonstrates a trend towards reduction of both Aβ40 and Aβ42 in ACI-24 vaccinated mice. Graphs are presented as mean ± SEM (n.s. non-significant, * *p* < 0.05, ** *p*< 0.001, unpaired two-tailed Student’s *t*-test).

**Figure 3 cells-12-00079-f003:**
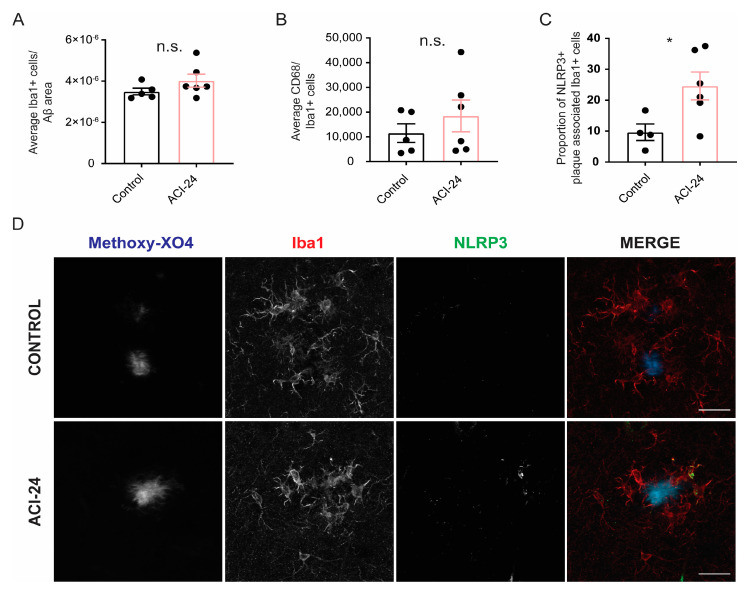
Increased microglial activation at Aβ plaques can be observed in the offspring generated by the transgenic mother following ACI-24 vaccination. Statistical analysis of the average number of microglia recruited at the Aβ plaques (plaque-associated microglia) of 5 control and 6 ACI-24 vaccinated mice (transgenic mother offspring) (**A**). The number of Iba1+ plaque associated microglia was normalised to the Aβ area (as stained with 3552 antibody). Average CD68 signal was measured and normalised to the microglia number, revealing a trend towards increased CD68 coverage at the Aβ plaques (**B**). Double staining of Iba1 and NLRP3 reveals an increase in the proportion of NLRP3+ microglia at the Aβ plaque (as stained with Methoxy-X04) following ACI-24 vaccination (**C**,**D**). Scale bar: 20 µm. Graphs are presented as mean ± SEM (n.s. non-significant, * *p* < 0.05, unpaired two-tailed Student’s *t*-test).

**Figure 4 cells-12-00079-f004:**
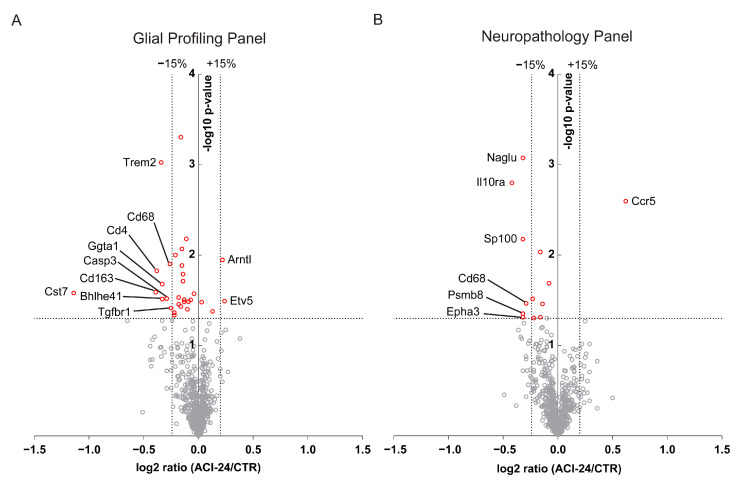
Bulk transcriptomic analysis from cerebrum of 6 ACI-24 vaccinated and 5 control mice (transgenic mother offspring). Volcano plots depict differentially expressed genes with the nCounter Glial Profiling Panel (**A**) and Neuropathology Panel (**B**). The negative log10 transformed *p*-value is plotted against the mean log2 transformed ratio between ACI-24 vaccinated and control animals. Horizontal dotted line depicts the *p*-value of 0.05, while the vertical lines represent changes in the expression of 15% as annotated. Genes with the *p*-value less than 0.05 are marked in red, while in grey are those with a *p*-value larger than 0.05. Differentially expressed genes that are significant according to the *p* value of 0.05 (non FDR-corrected) and changed more than 15% are marked with their names.

**Figure 5 cells-12-00079-f005:**
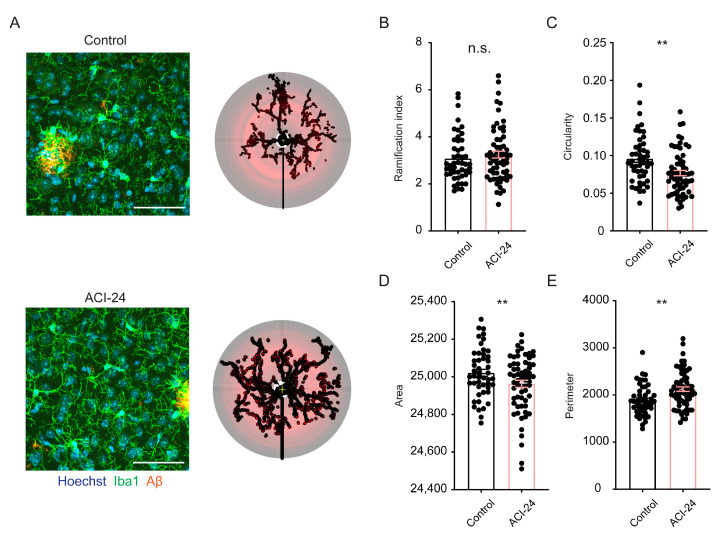
Plaque-distant microglia show increased ramification following ACI-24 vaccination. Representative images and the respective Sholl analysis of the plaque-distant microglia (8–14 microglia per brain) in 5 control (in total 48 microglia) and 6 ACI-24 (in total 63 microglia) treated mice (transgenic mother offspring) (**A**). Microglia are depicted with Iba1, Aβ is visualised using 3552 antibody and nuclei using Hoechst. Scale bar: 50 µm. Statistical analysis representing graphs for the ramification index (**B**), circularity (**C**), area (**D**) and perimeter (**E**) of Iba1+ cells. Graphs are presented as mean ± SEM (n.s. non-significant, ** *p* < 0.001). Kolmogorov–Smirnov test was performed to check if samples are following normal distribution, and then an unpaired two-tailed Student’s *t*-test or a Mann–Whitney test were carried out accordingly.

**Figure 6 cells-12-00079-f006:**
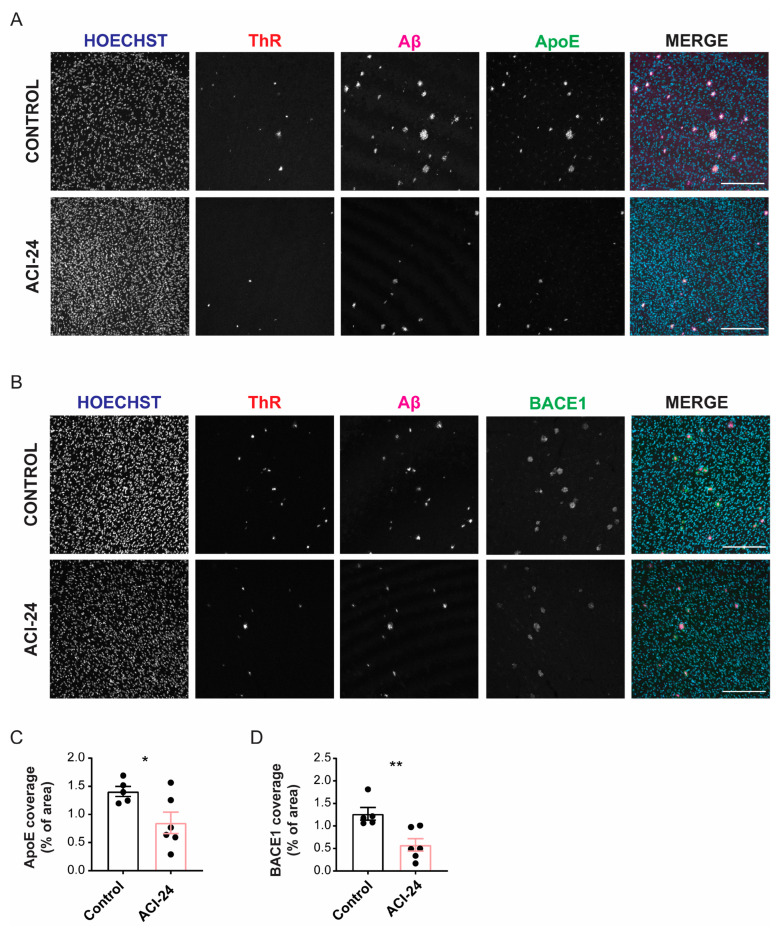
ACI-24 reduces ApoE protein levels and neuronal injury in the offspring generated by the transgenic mother. Representative images of control and ACI-24 vaccinated mice stained for nuclei (Hoechst), fibrillar Aβ (ThR), Aβ plaques (3552 antibody) and ApoE (**A**); and nuclei (Hoechst), fibrillar Aβ (ThR), Aβ plaques (NAB228 antibody) and BACE1 (dystrophic neurites) (**B**). Scale bar: 200 µm. Statistical analysis of 5 control and 6 ACI-24 vaccinated mice (transgenic mother offspring) showing significant downregulation of both total ApoE (**C**) and BACE1 (**D**) coverage. Graphs are presented as mean ± SEM (* *p* < 0.05, ** *p* < 0.001, unpaired two-tailed Student’s *t*-test).

## Data Availability

The mass spectrometry proteomics data have been deposited to the ProteomeXchange Consortium via the PRIDE partner repository with the dataset identifier PXD038665. Source data are provided with this manuscript.

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
