# Peer review of "Beneficial Effect of ACI-24 Vaccination on Aβ Plaque Pathology and Microglial Phenotypes in an Amyloidosis Mouse Model"

_cells, 2022, doi:10.3390/cells12010079_

Round 1

Reviewer 1 Report

This is a timely study related to the effects of AVI-24 vaccination (N-terminal Aβ fragments) on Abeta pathology and microglia phenotypes in an amyloidosis mouse model. ACI-24 is a liposome vaccine designed to elicit an antibody response against aggregated Aβ peptides without concomitant pro-inflammatory T cell activation. There are only minor points to be addressed:

- Are those differentially expressed microglial targets false discovery rate (FDR)-corrected indicted in the Table 1? Also, please add titles for the table 1 and 2.

- Authors used BACE1 staining to show plaque-associated dystrophic neurites (Fig 5). What about applying more frequently used markers to show alterations in dystrophic pathology, such as LAMP1?  

Reviewer 2 Report

Njavro et al characterise the effects of an antibody against amyloid-beta (Aβ), ACI-24, on amyloid plaques as well as associated microglia pathology using the APP/PS1 mouse model of cerebral amyloidosis. The authors show ACI-24 successfully reduces amyloid burden, specifically in offspring derived from female transgenic APP/PS1. The authors speculate in the discussion and in the abstract as to how this mechanism may influence neurodegenerative diseases such as Alzheimer’s disease (AD) and its pathology/pathogenesis. While it is reasonable to speculate about this in the Discussion, the manuscript does not conclusively demonstrate the effects of ACI-24 on microglia reactivity. The interpretation of results on microglia suffers from overinterpretation, especially as some of the data is contradictory or inconclusive due to the low power of the study. With this in mind, the reviewer submits the following comments:

Major concerns

1.     In Fig 5, the authors should show Aβ plaques costained with apoe and BACE1 to get a better representation of plaque-associated neuritic dystrophy. Similarly, the quantification needs to be normalized on a per plaque basis. Considering the authors insinuate in the discussion that plaque-associated damage is less in ACI-24 mice from transgenic mothers, potentially due to a reduction in fibrillar plaque, this data needs to be shown.

2.     Please explain the discrepancy in lack of microglial proteins changed in the proteomics data compared to significant changes in microglial genes highlighted in Table 1 and 2. The resolution of isolated microglia used for mass-spectrometry should be greater than that of bulk tissue used for nanostring analysis. Yet, the proteomics data show no significant changes in microglia proteins, including CD68 and Trem2, that the authors highlight as being significantly downregulated from bulk tissue.

3.     Please explain why the CD68 stain in Fig 3 shows an increase when the nanostring data shows a significant decrease by 17%, as highlighted by the authors in the main text. The authors discuss that an increase in CD68 and NLRP3 suggests an increase in recruitment of microglia to Aβ plaques potentially by FCGR-mediated phagocytosis, however their data is in direct contradiction to this statement. Please show that ACI-24 treatment increases phagocytic microglia similarly to their previous ex vivo studies cited in the manuscript.

4.     The authors state that ACI-24 treatment results in an overall reduced level of microglial activation status suggesting resolution of neuroinflammation, however the only results provided in favour of this argument are the changes in microglia morphology and NLRP3 reduction. The nanostring data is inconclusive due to the contradictory results on Cd68. Please provide additional immunohistochemistry data on changes in microglia markers such as Trem2, Clec7a and P2RY12 as well as cytokines and chemokines in the vaccinated vs. control mice.

Minor concerns

1.     Please provide representative of images from Aβ stainings from which Fig 2A-F were quantified.

2.     Similarly provide representative images for IBA1, CD68 and NLRP3 stainings in Fig 3.

3.     Considering the ramification index of microglia show no significant changes, the rest of microglial morphology paradigms (circularity, area and perimeter) do not necessarily indicate that these microglia are more ameboid. One would assume that the perimeter of ameboid microglia would be smaller rather than significantly larger as shown in Fig 4 E. The current data set would be strengthened by adding measurement of the cell soma diameter.

4.     Please show the nanostring data from Table 1 and 2 as volcano plots, or pi charts to get a better visual of the percentage of genes significantly changed from ACI-24 treatment vs. all genes included in the nanostring panel.

5.     Please rephrase  “clearance” to removal or reduction of  as the authors have not included any data that specifically addresses  clearance.

6.     One of the major issues for  immunotherapy is that of amyloid-related imaging abnormalities (ARIA). Does ACI-24 treatment lead to increased cerebral microhemorrhages? It would be helpful to quantify cerebral microbleeds in ACI-24 vs. control mice.

Reviewer 3 Report

This study tried to do some additional experiments to support the benefits of the old ACI-24 vaccine on AD-related pathology in the APPPS1 model. From the data presented, it is hard to see any encouraging readouts. There are also several obvious problems needing to be taken into consideration as follows.

1)      Abeta accumulation is an age-dependent process. It is not ideal to test the effects on very young mice.

2)      Was there a dose-dependent evaluation?

3)      Due to the large variation, only a limited number of mice were used in this study.

4)      From figure 2, the most convincing data (MSD) did not show any apparent difference between treatment and non-treatment although there is some trend.

5)      The bands in Figure 2G are separate, it is ideal to display the blots from all the mice due to the large variation.

6)      Where are the original data for Figure 3C? There is no significant difference in Figure 3A and 3B not as the authors stated in the text.

7)      Is there any change in the levels of Iba1 protein? How about the related astrocyte? GFAP+ morphology and GFAP levels?

8)      Figure 5 quality is not good. Did the authors measure their protein levels? Did the positive signaling (either APOE or BACE1) colocalize with Abeta plaques?

9)      Overall, as the authors stated, it displayed some effects on the APPPS1 mice with only lower Abeta levels, which means this potential drug in this study did only display very weak/mild and not encouraging effects.

Reviewer 4 Report

This study examined the effects of an Alzheimer's disease vaccine in a mouse model, demonstrating improvements in pathology.

Page 2, paragraph 1: Can you add a brief explanation in the text on what you mean by "functional repair" and "functional microglia"?

Page 2, paragraph 2: Change "pre-Aβ seeds" to "pre-amyloid seeds".

Page 3, paragraph 2: Change "within a palmitoylated lysine tandem" to "between palmitoylated lysine tandems".

Page 3, paragraph 2: Change "restores" to "prevents".

Page 3, paragraph 3: Change "AP-PPS1" to "APP-PS1" or "APPPS1". APP and PS1 are names of genes.

Part 2.1: Change "AP-PKM670/671NL" to "APPKM670/671NL".

Part 2.2: Change "12 weeks" to "10 weeks" since the last vaccination was at age 10 weeks.

Part 2.2: Change "Mice were re-genotyped" to "At age 12 weeks, mice were re-genotyped".

Part 2.2: It took me a while to understand "(5 female mice bred with mating scheme including transgenic APPPS1 father and the WT mother i.e. transgenic father offspring and 5 female mice bred with the opposite mating scheme – transgenic APPPS1 mother and WT father i.e. transgenic mother offspring)". Please rewrite to make it more clear.

Figures: Use higher resolution.

Figure 1B: I can't see the different colors or symbols of the 3 lines. Instead, you might use a solid, dashed, and dotted line. Make sure the legend clearly shows the 3 types of lines.

Figure 1C: The white boxes have black notches in them; explain these in the caption.

Figure 1: Change "image" to "images". Change "show" to "in cortical sections show". Change ". Data are analysed in the cortex (scale bar: 200 μm) and the boxed region is the higher magnification (scale bar: 50 μm)." to ", with scale bar for images in the center column indicating 200 μm, and scale bar for images in the right column (magnification of boxed regions from center column images) indicating 50 μm."

Figure 1D: I don't think you mean "% of Aβ coverage". I guess you mean "Aβ coverage (% of area)". Change the vertical axis label as appropriate.

Figure 1E: The vertical axis says "Plaque number". In what area is that number counted: μm squared? If so, state it on the axis label: Plaques (number/μm squared).

Figure 1: Change "is significantly downregulating" to "significantly downregulated".

2.3: Change "before (baseline), one week after each immunization, and at the end of the study" to "before the first immunization (baseline) and at one, three, and six weeks after the first immunization".

2.6: Add a comma to the sentence that starts "For Apoe antibody...".

2.7: Did the person handling quantification know the treatment group of each mouse being analyzed, or was the person blinded?

2.7: What is "the radius surpassing the longest branch of the cell"? That sounds like a length longer than any length of the cell: but how much longer, and why?

2.8: Change "β-actin" to "anti-β-actin", or something like that.

2.10: You may use a word different than "extraction" to distinguish this process from what you described in the previous section (2.9) as extraction. Maybe change "had been extracted" to "was removed from the skull" or something like that.

2.12: Capitalize "Independent Acquisition".

2.12: Spell out what LFQ stands for the first time you use it.

Figure S2A. Please start the vertical axis at 0, not 1.

Figure 3. Change "a trends" to "a trend".

3.5: Change "several of DAM genes" to "several DAM genes".

Figure 5B: Change "% of ApoE coverage" (which means that you detected around 0.5 to 1.5 percent of the total ApoE coverage) to something like "ApoE coverage (%)" (which is what you intended to mean).

Figure 5C: Change "% of BACE1 coverage" to "BACE1 coverage (%)".

4: Change "antibody titers directed against the Aβ1-42" to "Aβ1-42 antibody titers", or something like that.

4, paragraph 1, last sentence: The meaning would be very different depending on whether there is a comma before "that". Without the comma, the sentence means that vaccination had more effect in a subset of the animals with transgenic mothers (particularly, the subset with lower Aβ burden than that of animals with transgenic fathers) than in other animals. With a comma, the sentence means that vaccination had more effect in the animals with transgenic mothers (not only in a subset of them) than in animals with transgenic fathers. I think you intended to mean the latter, thus you should add a comma. But if you intended the former, you should show the statistics supporting your claim in the Results. In a scientific paper--like in a legal contract, an engineering plan, or a medical prescription--punctuation can make a critical difference; please be careful!

4, paragraph 2, first sentence: Say either "This unexpected difference" or "These unexpected differences". Change "observed in" to "between". 

4, paragraph 2: The hypotheses you mention in this paragraph are interesting and worthy of further exploration, but you should also mention one more possible explanation, which is simply by chance. You might further speculate that if the maternal transfer of antibodies is the mechanism, that may imply that vaccination should be given even earlier, to attempt to approach the success of the maternally transfered antibodies.

4, paragraph 4: "less severe side effects such as ARIA"--less severe compared to what? In the paper, please clarify the point you are making here.

4, paragraph 4: Early in the paragraph you note "reduction in the microglial activation state", but later you "suggest local microglial activation". These seem contradictory. I suppose there might be local activation followed by global quiescence, but you can explain in the paper.

Round 2

Reviewer 3 Report

The endeavors made by the authors to improve the MS were appreciated and satisfied.